# Comparing a New Non-Invasive Vineyard Yield Estimation Approach Based on Image Analysis with Manual Sample-Based Methods

Gonçalo Victorino [1,2], Ricardo P. Braga [1] , José Santos-Victor [2] and Carlos M. Lopes [1,*]

1 Linking Landscape, Environment, Agriculture and Food (LEAF), Instituto Superior de Agronomia, Universidade de Lisboa, Tapada da Ajuda, 1349-017 Lisbon, Portugal; gvictorino@isa.ulisboa.pt (G.V.); ricardobraga@isa.ulisboa.pt (R.P.B.)
2 Institute for Systems and Robotics-Lisboa (ISR), Instituto Superior Técnico, Universidade de Lisboa, Avenida Rovisco Pais, 1049-001 Lisbon, Portugal; jasv@isr.tecnico.ulisboa.pt
* Correspondence: carlosmlopes@isa.ulisboa.pt

**Abstract:** Manual vineyard yield estimation approaches are easy to use and can provide relevant information at early stages of plant development. However, such methods are subject to spatial and temporal variability as they are sample-based and dependent on historical data. The present work aims at comparing the accuracy of a new non-invasive and multicultivar, image-based yield estimation approach with a manual method. Non-disturbed grapevine images were collected from six cultivars, at three vineyard plots in Portugal, at the very beginning of veraison, in a total of 213 images. A stepwise regression model was used to select the most appropriate set of variables to predict the yield. A combination of derived variables was obtained that included visible bunch area, estimated total bunch area, perimeter, visible berry number and bunch compactness. The model achieved an $R^2 = 0.86$ on the validation set. The image-based yield estimates outperformed manual ones on five out of six cultivar data sets, with most estimates achieving absolute errors below 10%. Higher errors were observed on vines with denser canopies. The studied approach has the potential to be fully automated and used across whole vineyards while being able to surpass most bunch occlusions by leaves.

**Keywords:** grapevine yield prediction; bunch occlusion; proximal sensing; grape pixels; bunch compactness

## 1. Introduction

The capacity to accurately estimate the yield of a vineyard in the early stages of development can be extremely advantageous for the winegrower, with impacts on the whole vine and wine production chain. In the vineyard, it can help plan strategies of grape thinning as a yield regulation practice or to plan harvest in terms of logistics, scheduling, workforce and machinery requirements [1,2]. In the cellar, it helps planning for tank space allocation, crusher intake scheduling and oenological products. Furthermore, it provides advantages in terms of marketing and wine stock management [3]. Current conventional methods of yield estimation consist of manually sampling grapevine yield components using randomized vine segments in the field [4]. This sampling can be performed at various phenological stages, and yield potential can be estimated using historical data (e.g., average bunch weight at harvest). Early forecasts are important and can help the winegrower adopt strategies to adjust their production in that season [5]. However, if performed too early, yield estimation can be inaccurate as there are numerous factors, such as climatic events and other abiotic and biotic stresses, that can affect yield components after the estimation date. Therefore, conventional yield estimates are usually performed close to the veraison phenological stage (BBCH 81; [6]) and generalized for the whole vineyard plot [7]. This method, described in Equation (1), is easy to perform and independent of any technology,

as it is based only on historical bunch weight and on bunch counts, a yield component that can explain, on average, about 60% of the total yield variation [8]. It allows for early estimations, as bunch counts can be performed before fruit set. However, the dependency on historical data regarding bunch weight at harvest is a disadvantage of this method. An alternative manual method consists of weighing a sample of bunches at the lag phase (just before veraison) while considering a growth factor until full maturation (Equation (2)). Berry lag phase is the phenological stage just before the onset of the veraison stage (between BBCH 79 and 81), corresponding to the phase where berry growth slows down as seeds begin to harden. Either the lag phase or the veraison phenological stage are stages not far from harvest, presenting a lower risk of unpredictable events but early enough to allow the farmer to make use of most adaptation strategies mentioned above. However, both of these manual methods are dependent on manual counting or sampling in the field, making them sensitive to spatial variability. These tasks require a significant amount of labor, especially in highly variable vineyards where a higher number of samples is required for an accurate yield estimation [9,10]. Moreover, the lag phase method, apart from being based on destructive samples, is dependent on a historical berry growth factor, which is largely affected by growth conditions, cultivar and management practices [11–13].

$$\text{Yield} \left( \frac{\text{kg}}{\text{ha}} \right) = \frac{\text{N}^{\text{o}} \text{ bunches}}{\text{vine}} \times \text{Bunch weight harvest (kg)} \times \frac{\text{N}^{\text{o}} \text{ vines}}{\text{ha}} \tag{1}$$

$$\text{Yield} \left( \frac{\text{kg}}{\text{ha}} \right) = \frac{\text{N}^{\text{o}} \text{ bunches}}{\text{vine}} \times \text{Bunch weight lag phase (kg)} \times \text{Growth fact} \times \frac{\text{N}^{\text{o}} \text{ vines}}{\text{ha}} \tag{2}$$

where the growth factor consists of the ratio between bunch weight at the lag phase and at harvest (historical data). Manual yield estimation methods are known to have an average accuracy error between 10% and 40% relative to real yield at harvest [8,14–16]. Some authors consider an error ranging between 5% [4] and 10% [15] as ideal.

Considering the disadvantages of the manual methods regarding their cost, possible destructiveness and unreliable estimation error, recently, several efforts have been made to replace them with automatic non-invasive approaches. Several alternatives to manual vineyard yield estimation have been recently reported [17], for example, based on the use of vegetation indexes obtained via aerial imaging [18], airborne pollen samples obtained using Crus pollen traps [19,20], trellis tension sensors deployed across the vineyard [21], crop simulation models using environmental and physiological data [22] and sensor technology such as radar or ultrasonic for bunch detection in dense canopies [23,24]. Overall, all methods present advantages and disadvantages when compared to manual approaches. While all are independent of bunch sampling, some are limited to region-wide estimations (e.g., airborne pollen). Others are dependent on well-maintained trellis systems and on a good distribution of sensors across the whole vineyard (e.g., trellis tension). Others depend on vegetative parameters that may not always be related to yield as well as expert-dependent technology that is not always readily available (e.g., sensor-based methods).

Out of the mentioned alternative yield estimation approaches, the most explored ones in recent research have been data driven-models based on computer vision and image processing [17]. Such approaches, besides being non-invasive, by scanning extensive amounts of individual plants, can reduce the errors related to spatial and temporal variability while also collecting information regarding their geographical position.

The most common computer vision-based approaches consider the automatic recognition of either bunch overall pixels or single berries as they represent the main estimators of yield in 2D image scenarios [25]. Image analysis has been used to accurately identify the number of visible berries in vine images, using supervised learning algorithms [26]. Different approaches have been explored to identify the same yield component, such as hedge detection using Hough transform [27], Boolean models [28] and random forests [29]. Average berry size has also been automatically estimated using computer vision techniques [30,31]. Recently, grape bunch projected area has been detected with an unmanned aerial vehicle using photogrammetric point clouds and color filtering [32]. Other works explored the automatic detection of bunches on ground images using support vector ma-

chines [33] and artificial neural networks [34]. Furthermore, 3D imaging has also been explored for bunch detection (e.g., [35,36]), where it has been explored as only outperforming 2D analysis in laboratory conditions.

Overall, yield components automatic recognition in vine images has proven successful in the above-referenced works, as most of them focus on the effectiveness of the automatic algorithms and their accuracy in identifying key vine image features that allow estimations of yield. In fact, several referenced works also perform yield estimations (e.g., [27–29,36]); however, apart from early-stage approaches (e.g., [37]), the majority of these research works apply their methods after some canopy manipulation. They do it by defoliating vines at the bunch zone prior to image collection (e.g., [3,33,34]) to surpass one of the main challenges of yield estimation via image analysis: bunch occlusion by leaves. In fact, most research does not account for the yield components that are occluded by leaves, woody material and neighboring bunches, previously referred to as vine-occlusions and self or cluster-occlusions [38]. Furthermore, vineyards with very dense canopies are not usually considered. In such cases, only a small fraction of bunches is visible and not necessarily correlated with vine yield [39], and defoliation is not always a common practice (e.g., warm climate viticulture). The visibility of yield components along the vine growing cycle has been previously explored [40], and bunch occlusion ratios between 50% and 75% at the veraison stage were observed. Artificially generated vine images were recently explored to develop an algorithm capable of estimating berry numbers behind the leaves [41]. This work showed promising results in their synthetic data but inconclusive in real vine images. In recent research, our team showed that bunch occlusion by leaves could be surpassed by considering other canopy traits, besides yield components, such as canopy porosity [42]. Previously, canopy porosity has been estimated via image analysis [43] and presents results similar to the ones obtained via classical approaches [44]. Visible bunch area and canopy porosity have been used to estimate the percentage of exposed bunches [42]. This percentage was then used to compute the total bunch area, which, in turn, was used as a proxy of vine yield by means of a simplified area-to-mass conversion.

After yield components recognition and estimation considering their occlusion, it is necessary to convert that information into vine yield, which is a challenging step as it can be highly cultivar-dependent [25,45]. Bunch morphology can vary greatly among cultivars and even between sites for the same cultivar [46]. The bunch compactness trait has been explored, considering its variability between cultivars and its importance regarding grape quality and bunch health [47,48]. Our research team also demonstrated that bunch morphology is a key feature when attempting to achieve a multicultivar bunch weight estimation model [45]. Furthermore, we also proved that when converting bunch image features into weight, several image-based traits (e.g., visible berries, bunch area, average berry size) from one single bunch image are not redundant and present better results than when using single predictors on bunch images [25,45].

Today, the advantages that an image-based yield estimation approach has, compared to a manual one, are clear, especially considering its potential to be completely non-destructive, faster and labor free. However, its estimation errors in real field conditions, with high magnitudes of occlusion, are still difficult to be accepted by the growers. The aim of this paper is to compare the accuracy of a non-invasive and multicultivar, image-based yield estimation approach, in real field conditions, with a manual method based on bunch counts and historical bunch weight at harvest.

## 2. Materials and Methods

### 2.1. Data sets, Plant Material and Growth Conditions

Data used in this paper were collected from three vineyard plots located at two sites within the Lisbon wine region, approximately 50 km apart (Figure 1). The first site, Instituto Superior de Agronomia (ISA; 38°42′24.61″ N 9°11′05.53″ W), located at the ISA campus, encompasses two vineyard plots located at approximately 50 m above sea level, trained in a vertical shoot positioning trellis system with two pairs of movable wires. The first vineyard

plot consists of drip-irrigated spur pruned white cultivars Alvarinho, Arinto and Encruzado grafted onto 1103 Paulsen rootstock, planted in 2006, spaced 1.0 m within and 2.5 m between north-south oriented rows. Water was supplied with a drip irrigation system, and irrigation was managed using a soil water probe. Readily available water was maintained until veraison, at which point water was applied by replacing only 50% of crop evapotranspiration. The second vineyard plot of site ISA consists of rainfed spur pruned Syrah cultivar grafted onto 140 Ruggery rootstocks, planted in 1999, spaced 1.2 m within and 2.5 m between north-south oriented rows. At this vineyard, data were collected from the 2019 season. In both plots, the soil is a clay loam with approximately 1.6% organic matter and a pH of 7.8.

The second site, Quinta da Amieira (QA), is located at Cabeça Gorda, approximately 60 km north of Lisbon (39°11′26.442″ N 9°16′2.62″ W), 100 m above sea level. The vineyard was planted in 2003, spaced 1.0 m within and 2.5 m between north-south oriented rows, grafted onto SO4 rootstock, trained in a vertical shoot position, and supported by two pairs of movable wires. At this vineyard, data were collected from the cultivars Castelão (red) and Chardonnay (white) in 2020. Site QA is a rainfed vineyard and presents a sandy clay loam soil with 1.5% organic matter and a pH of 5.9.

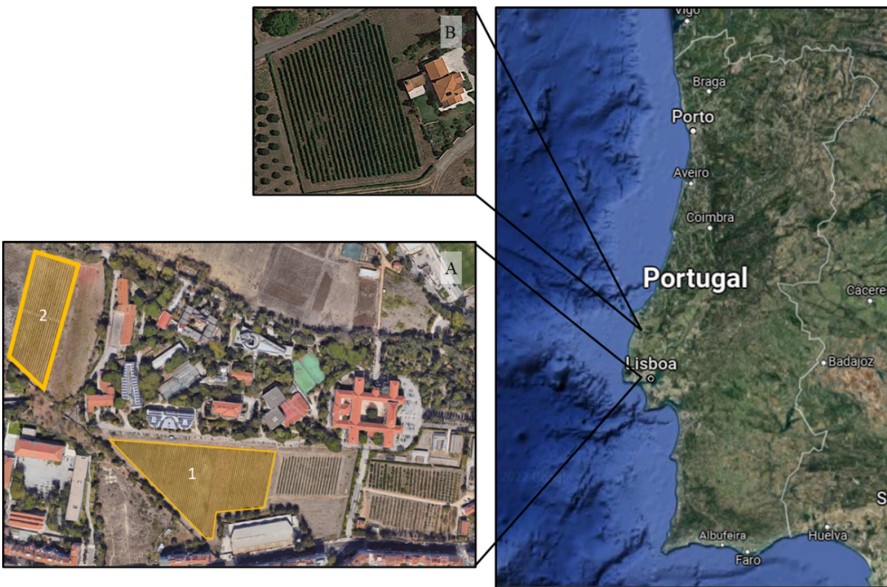

**Figure 1.** Location of studied sites (A = ISA; B = QA) and respective vineyard plots (1 and 2) in the Lisbon winegrowing region, Portugal. Image acquired from Google Earth®®®.

Both sites present a typical Mediterranean climate with an Atlantic influence. Climatic data were collected from a weather station located at ISA and one other approximately 10 km away from the QA site. The 2019 season at ISA presented a spring with an average of ~119 mm of precipitation from March to June, followed by a dry and warm summer (~17 mm of precipitation and an average mean temperature of 20.6 °C, from June to September). The 2020 season at QA presented a drier spring with an average of ~53 mm of precipitation from March to June and warm and dry summer (~28 mm of precipitation and an average mean temperature of 21.3 °C from June to September). Whereas the 2021 season at ISA presented a spring with ~103 mm of precipitation (March until June) and a very dry summer with ~10 mm from June until September. In all studied seasons, both sites were subject to similar standard cultural practices during the growing cycle, including water shoot removal, shoot trimming at 1.2 m above the cordon, shoot positioning and fertilization.

In total, the present experience used six subsets of data with different cultivar, season and site combinations (Table 1). All data were collected on 1 m length vine segments near the veraison phenological stage (BBCH 81; [6]). A total of 213 vine segments were analyzed (30–40 per data set).

**Table 1.** Characterization of each data set. Static image type was collected with a commercial camera, and on-the-go images were collected using a terrestrial robot platform (Vinbot; [49]).

| Acronym | Cultivar | Site | Season | Image Type | N | Vine Spacing | Plot Size (Plant Number) |
|---|---|---|---|---|---|---|---|
| Al21 | Alvarinho | ISA | 2021 | Static | 36 | 1.0 × 2.5 m | 1000 |
| Ar21 | Arinto | ISA | 2021 | Static | 32 | 1.0 × 2.5 m | 1040 |
| Ca20 | Castelão | QA | 2020 | On-the-go | 30 | 1.0 × 2.5 m | 137 |
| Ch20 | Chardonnay | QA | 2020 | On-the-go | 36 | 1.0 × 2.5 m | 139 |
| Enc19 | Encruzado | ISA | 2019 | On-the-go | 39 | 1.0 × 2.5 m | 1065 |
| Sy19 | Syrah | ISA | 2019 | On-the-go | 40 | 1.2 × 2.5 m | 670 |

*2.2. Image Acquisition*

As mentioned in Table 1, the image collection setup varied throughout the trial (Table 1). As image analysis was performed manually, this was irrelevant to the quality of data collection since actual, as opposed to estimated, parameters were obtained. Images were collected using either a commercial camera, statically, or with the ground platform Vinbot [49] on-the-go. The commercial camera model used was a Nikon D5200 (Nikon Inc., Melville, NY, USA), with a resolution of 6000 × 4000 pixels and a Sigma 50 mm F2.8 macro lens. The Vinbot uses a Kinect v2.0 (Microsoft Corp., Albuquerque, NM, USA) with a resolution of 1920 × 1080 pixels. Along with the camera, the static image acquisition system encompassed a tripod that was placed approximately 2 m away from the target vine and held the camera approximately 1 m above the ground. Furthermore, the on-the-go image system involved using the platform Vinbot, an unmanned platform that carries several sensors for vineyard scanning [49]. In this experiment, only the robot's RGB camera was used. To facilitate image analysis, a blue background was used on both image acquisition systems. Such an approach made it easier to distinguish vine elements from background noise and to estimate canopy porosity. All vine segments presented a static plastic scale under the cordon to identify the segment and to provide scaling information for image analysis. To avoid direct light incidence and consequent poor-quality imaging, images were collected after midday, taking advantage of the north/south row orientation and of the shade provided by the background positioned on the opposite side of the canopy combined with the fully developed canopy. Figure 2 presents one example for each of the datasets described above.

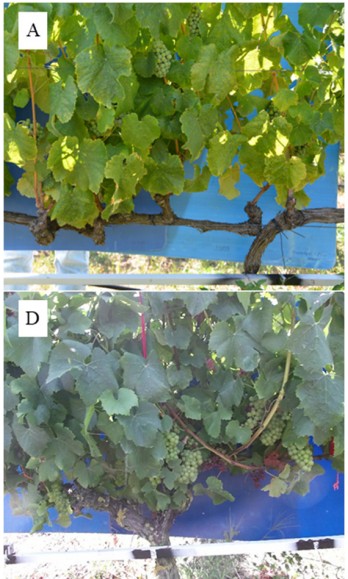
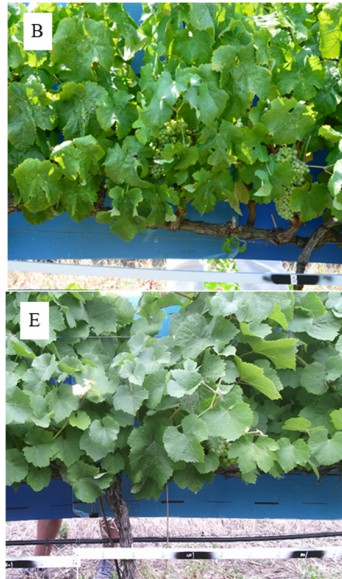
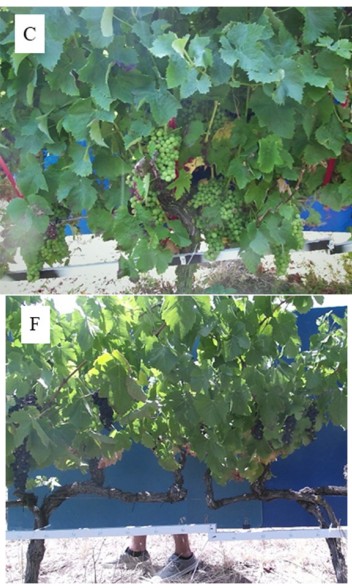

**Figure 2.** Examples of images collected for each cultivar of the data set. (**A**): Alvarinho ISA 2021; (**B**): Arinto ISA 2021; (**C**): Castelão QA 2020; (**D**): Chardonnay QA 2020; (**E**): Encruzado ISA 2019; (**F**): Syrah ISA 2019.

### 2.3. Image Processing

From each vine segment, the following variables were extracted: visible bunch projected area (vBA), visible bunch projected perimeter (vBP), visible berry number (vBe) and canopy porosity (POR). To assess these features, all images were processed as follows:

1.  Images were scaled based on the physical scale present under the cordon of each vine segment;
2.  Images were cropped in order to include only a region of interest that encompasses approximately 50 cm above the cordon (Figure 3);
3.  Bunches were manually segmented on each image in order to obtain the actual vBA and vBP from each vine segment;
4.  The vBe was manually counted on each image;
5.  To estimate POR, canopy gap pixels were classified using a static color threshold after converting the original image from RGB to HSV (Hue-Saturation-Value) color space for improved invariance to illumination conditions. This task was performed using OpenCV in PyCharm®®®, an open-source Python-focused IDE, for automation purposes.

Tasks 1 to 4 were performed with the software Image J®®® (v1.53k, National Institutes of Health, EUA). Image features were selected based on previous research [42,45].

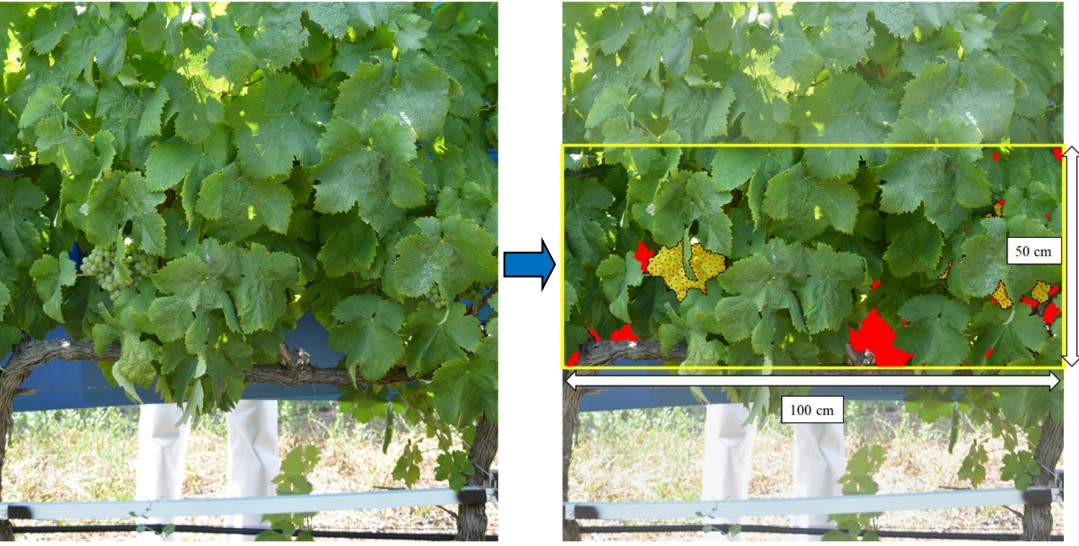

**Figure 3.** Example of an analyzed image obtained from the cultivar Arinto and respective region of interest for image analysis and feature segmentation. Red-filled polygons correspond to canopy porosity; yellow-filled polygons + black dashed contour indicates the visible bunch projected area + perimeter and red dots the visible berries.

### 2.4. Yield Estimation

All analyzed vine segments were harvested at full maturation, and their bunches were counted and weighed. The final yield was compared to the output of two different methods of yield estimation: (i) the manual method, based on bunch counts, conventionally used in commercial vineyards and thoroughly described as the traditional method in [7]; (ii) an image analysis method following recent research [42], followed by a conversion of bunch area into yield based on several yield-predicting vine traits. The image analysis method described in Victorino et al. [42] is based on the assumption that a relationship between canopy porosity and bunch exposure exists. The method uses a combination of canopy porosity and visible bunch pixels to estimate the occluded fraction of the bunch area.

### 2.4.1. Manual Yield Estimation

Manual yield estimation was performed according to Equation (1), adapted from [7]. This method relies on multiplying the number of bunches by the average bunch weight at harvest, based on historical data. Bunches were counted on all analyzed vines.

### 2.4.2. Yield Estimation via Image Analysis

Yield estimation via image analysis was obtained according to two different approaches, both based on a previously reported methodology [42]. In that methodology, vBA and POR were used to estimate the fraction of bunches occluded by leaves and, consequently, the total bunch projected area (TBA). In that paper, the authors have simplified the final step of converting TBA into weight by using a single factor of conversion, different for each studied cultivar. In the present work, TBA was considered along with other variables related to bunch weight [45]: vBP and vBe. However, these variables, unlike TBA, which already considers POR, are too dependent on bunch visibility and canopy density, which can be highly random. As such, they were only used to obtain derived variables, i.e., ratios between one another (e.g., vBA/vBe; vBA/vBP × vBe). Moreover, in an attempt to further enhance the model to be generalizable to different cultivars, a bunch compactness index (CI) was added. The CI used was based on the OIV descriptor nº 204 [46], which classifies bunches into five categories, based on the mobility of the berries and the visibility of the pedicels, namely: very loose (class 1), loose (class 3), medium (class 5), dense (class 7) and very dense (class 9). The CI, as it is a visual assessment, was not obtained from all images but instead only on a sample, with the aim to characterize the studied cultivar.

A flowchart is shown in Figure 4 illustrating the pre-selection of variables followed by a stepwise regression model computed for final vine yield estimation. As explained above, it relies on two inputs of data: (i) user input regarding CI and (ii) image features originating from image analysis, used to compute TBA and other derived variables. All variables were then used to feed the model for yield estimation for each vine segment.

### 2.5. Data Analysis

All cultivars' datasets were grouped together into one original dataset, which was divided into a training set (with 50% of the data) and a validation set (with the remaining 50%) in a fully random way. Data analysis can be divided into three steps. The first step consists of characterizing the main estimators of yield considered in each of the two yield estimation approaches used in this work, on the full data set, separated by cultivar. The second step involves the model computation for yield estimation using image analysis. For that, a forward stepwise regression model was fitted on the training set after performing a collinearity test. A 0.15 critical F statistic was used in order to select the most meaningful variables to estimate vine yield. The resulting multiple linear regression model was compared to other types of models such as support vector machines, decision trees and neural networks to explore non-linear possibilities. However, none presented clear advantages towards the linear regression model. The model was then validated on the validation set, and the coefficient of determination ($R^2$) and root mean squared error (RMSE) were used to evaluate its performance. Furthermore, an F-test for slope = 1 and for intercept = 0 was performed for overall model evaluation on both the training and validation sets. The third and final step consisted of comparing the output of the two above-mentioned yield estimation approaches. For this, to allow for a fair comparison, both approaches were used only on the validation data set, and estimation errors and trends were compared. The main error indicator was calculated as the difference between estimated and actual yield, divided by actual yield. The absolute value was not used in order to distinguish under from overestimations. The majority of the data analysis was performed using R Studio (v1.4.17; R Studio Inc.), while the stepwise regression model was computed using SAS<sup>®®®</sup> (v9.3; SAS Institute, Cary, NC, USA).

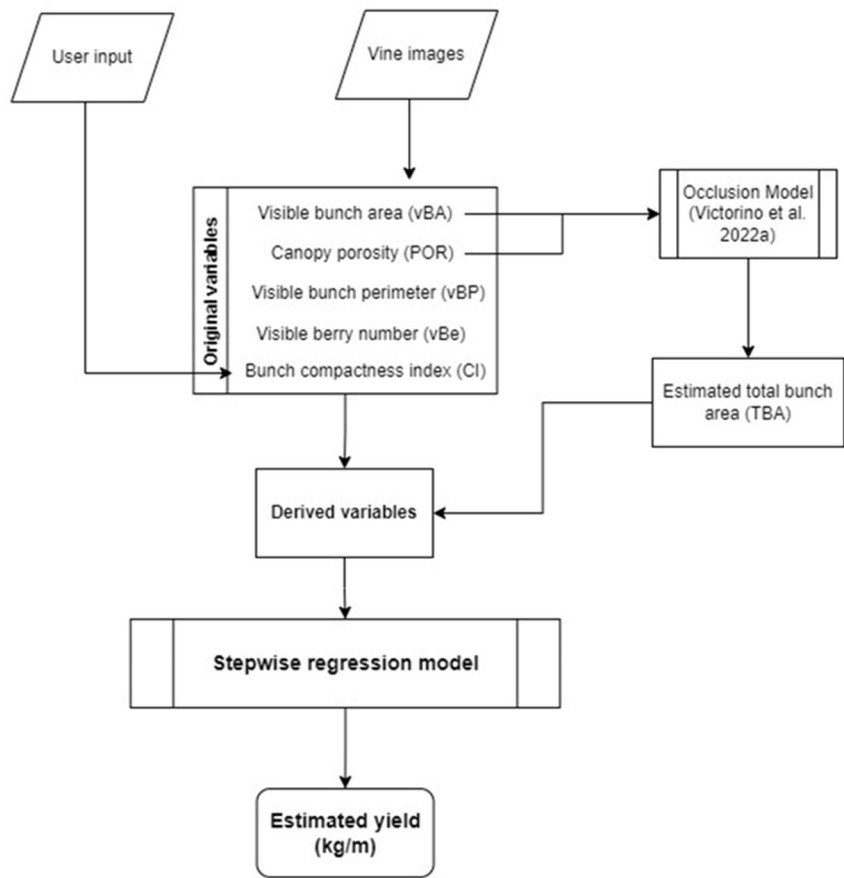

**Figure 4.** Flowchart describing the proposed yield estimation approach from image collection to modeling and final estimation [42].

## 3. Results

### 3.1. Yield Estimators

#### 3.1.1. Manual Sampling

The manual yield estimation approach explored in this work is based on one of the many possibilities reported in the literature, depending on bunch counts performed during the period of fruit set-veraison [7]. The actual bunch number was multiplied by their average historical weight at harvest (Equation (1)). Table 2 shows the average values for both yield components of the studied season plus historical bunch weight from five historical seasons, as well as the Pearson correlation coefficient between bunch number and actual yield, observed for each cultivar separately.

**Table 2.** Average ± standard error for the yield components used to obtain yield estimations using the manual method. Pearson correlation coefficient (r) between bunch number and actual yield. Data from the full data set, separated by cultivar. Bunch weight number of observations = ~50 independent bunches per case. Historical bunch weight was obtained from 5 historical seasons.

| Cultivar | n | Historical Bunch Weight (g) | Bunch Weight (g) | Bunches/m | Actual Yield (kg/m) | r |
|---|---|---|---|---|---|---|
| Al21 | 36 | 115.3 ± 11.8 | 84.2 ± 3.4 | 21.1 ± 1.0 | 1.9 ± 0.1 | 0.82 |
| Ar21 | 32 | 337.2 ± 32.0 | 410.5 ± 12.3 | 6.2 ± 0.5 | 2.4 ± 0.2 | 0.81 |
| Ca20 | 30 | 282.8 ± 25.8 | 244.7 ± 5.3 | 26.6 ± 1.4 | 6.5 ± 0.4 | 0.79 |
| Ch20 | 36 | 142.4 ± 18.5 | 118.3 ± 5.9 | 29.1 ± 1.4 | 2.1 ± 0.2 | 0.83 |
| En19 | 39 | 176.7 ± 17.0 | 170.4 ± 11.7 | 12.9 ± 0.7 | 1.8 ± 0.1 | 0.75 |
| Sy19 | 40 | 119.5 ± 13.3 | 72.1 ± 4.7 | 15.3 ± 0.8 | 0.8 ± 0.1 | 0.86 |

The heaviest bunches were found on cv. Arinto, followed by cvs. Castelão and Encruzado, while cvs. Alvarinho and Syrah presented the lightest ones. Historically, cv.

Arinto presented lighter bunches than in the present season. However, these were still the heaviest among all cultivars. All other cultivars presented higher historical bunch weights than in the present season, particularly cv. Syrah, on which historical bunch weight was almost twofold. The cv. Chardonnay presented the largest bunch number when compared to all other cultivars, closely followed by cvs. Castelão and Alvarinho. Cultivar Arinto showed the lowest number of bunches, by far. The correlation coefficients between bunch number and actual yield were highest for cv. Syrah, closely followed by cvs. Chardonnay and Alvarinho and lowest for cvs. Encruzado and Castelão.

3.1.2. Image Analysis

On the same plants where bunches were counted, images of the full vine segments were collected. Table 3 shows the average values of the variables obtained from these images. The table also includes TBA, which was estimated from vBA and POR, following recently developed methodologies [42], and CI, obtained from visual assessment of a sample of images.

**Table 3.** Average $\pm$ standard error for the variables used to obtain yield estimations using the image analysis method (Equation (3)). Data from the full data set, separated by cultivar. vBA: Visible bunch projected area; POR: Canopy porosity; vBP: Visible bunch projected perimeter; vBe: Visible berry number; TBA: estimated total bunch projected area ([42]); CI: Compactness index at veraison, determined using the OIV scale ([46]).

| Cultivar | N | vBA (cm$^2$) | vBP (cm) | POR (cm$^2$) | vBe | TBA (cm$^2$/m) | CI * |
|---|---|---|---|---|---|---|---|
| Al21 | 36 | 173.6 $\pm$ 10.1 | 193.1 $\pm$ 10.7 | 337.2 $\pm$ 52.6 | 193.1 $\pm$ 11.0 | 784.4 $\pm$ 29.8 | 5.0 |
| Ar21 | 32 | 123.4 $\pm$ 16.2 | 106.9 $\pm$ 10.6 | 162.1 $\pm$ 31.0 | 119.8 $\pm$ 15.3 | 807.9 $\pm$ 58.2 | 5.0 |
| Ca20 | 30 | 533.5 $\pm$ 40.4 | 333.1 $\pm$ 23.2 | 192.4 $\pm$ 19.3 | 269.0 $\pm$ 22.5 | 1128.1 $\pm$ 46.3 | 7.0 |
| Ch20 | 36 | 510.0 $\pm$ 39.1 | 349.8 $\pm$ 24.8 | 266.6 $\pm$ 39.6 | 296.5 $\pm$ 23.9 | 813.5 $\pm$ 39.4 | 5.0 |
| En19 | 39 | 85.3 $\pm$ 8.7 | 69.0 $\pm$ 5.6 | 112.0 $\pm$ 14.4 | 52.1 $\pm$ 5.5 | 638.5 $\pm$ 33.1 | 5.0 |
| Sy19 | 40 | 142.7 $\pm$ 12.6 | 124.8 $\pm$ 9.6 | 434.0 $\pm$ 49.4 | 209.3 $\pm$ 18.6 | 541.0 $\pm$ 29.5 | 3.0 |

* CI is fixed per cultivar and does not present deviation.

The highest vBA was observed on cv. Castelão, closely followed by cv. Chardonnay, which presented values three to five-fold higher than the remaining cultivars, with cv. Encruzado presenting the lowest average value. vBP presented relative differences that mirrored those reported for vBA. POR was highest in cv. Syrah, followed closely by cv. Alvarinho, with cv. Encruzado presenting the lowest values. vBe was highest in cv. Chardonnay, closely followed by cvs. Castelão and Syrah, with the lowest value being observed in cv. Encruzado. Regarding TBA, cv. Castelão presented the highest value, followed by Arinto, while the lowest values were observed on cvs. Syrah and Encruzado. Regarding bunch compactness, overall, cultivars were considered to have medium compact bunches at veraison, within the OIV scale [46], with the exceptions of cvs. Castelão and Syrah, which were classified as having dense and loose bunches, respectively.

Figure 5 shows the Pearson correlation coefficients between estimated TBA and actual yield, separated by cultivar (A), as well as the regression analysis between the same variables (B). When analyzed per cultivar, a high and significant correlation coefficient (r) was observed in all cases, with the lowest r being shown by cv. Encruzado and the highest by cv. Castelão. Considering the regression analysis (Figure 5B), a significant coefficient of determination was found between these two variables, which, overall, presented an exponential relationship. Some variability was found in the residuals of the regression model, particularly between cultivars. While almost all observations of cv. Castelão are above the regression line, in cv. Syrah, the opposite occurs. A covariance analysis corroborates the visual appraisal, indicating significant differences between cultivars in the relationship between TBA and actual yield (results not shown). Apart from this cultivar dependency, the model explains only 62% of the actual yield variability. Overall, the results indicate that the conversion of TBA into bunch mass requires further improvements, particularly considering its generalizing capability [42].

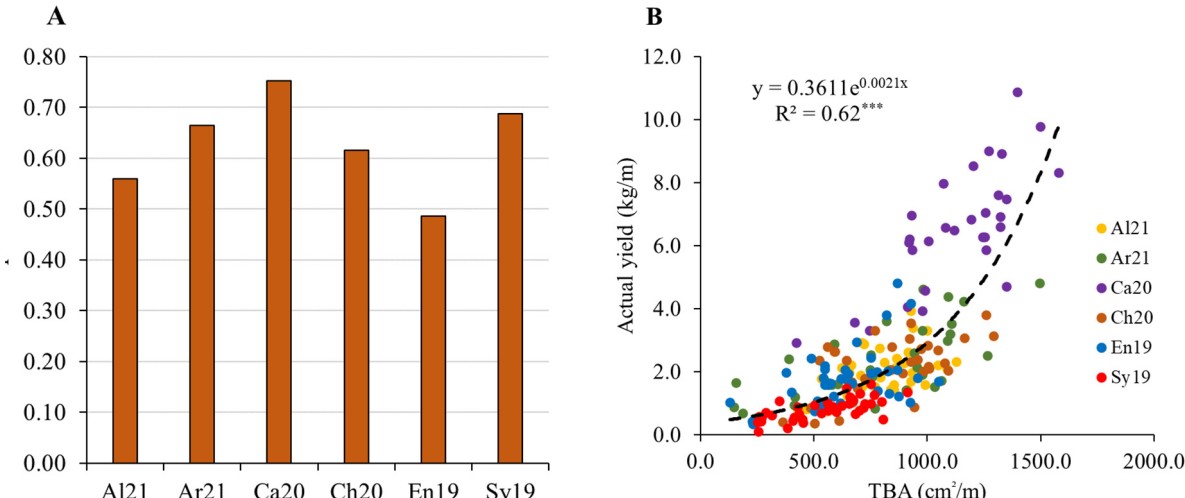

**Figure 5.** (**A**) Pearson correlation coefficients (r) between estimated total bunch area (TBA) and actual yield, separated by cultivar. (**B**) Regression analysis between estimated TBA and actual yield. *** indicates statistical significance ($p < 0.001$). Data from full data set (n = 213).

### 3.2. Improving the Conversion of Image Traits into Mass

In order to improve the accuracy of the TBA conversion into mass, a new model was explored. For that, apart from TBA, other image-based derived variables were computed for including in the stepwise regression. As described in the Materials and Methods section, the derived variables were calculated as the ratio or product between traits obtained directly from the vine images. These derived variables provide hints regarding bunch and berry architecture, bunch compactness and distribution in the image. Variable POR was left out as it was already included in the estimation TBA. Furthermore, in an attempt to further improve model performance, the variable CI was included.

To find an appropriate set of independent variables to predict yield, a forward stepwise regression analysis between all the above-mentioned variables and actual yield was performed. Table 4 shows the summary of the stepwise regression.

**Table 4.** Summary of stepwise regression used to estimate vine yield with respective added and removed variables, statistical significance for their selection (*p*-value), partial $R^2$ and MSE (mean squared error).

| Step | Added Variable | Removed Variable | *p*-Value | Partial $R^2$ | MSE (kg) |
|------|---------------|------------------|-----------|---------------|----------|
| 1 | TBA $\times$ CI$^2$ | - | 0.000 | 0.837 | 0.580 |
| 2 | vBP/Vba $\times$ vBe | - | 0.051 | 0.849 | 0.544 |
| 3 | vBA/vBP | - | 0.125 | 0.852 | 0.537 |
| 4 | vBA/vBP $\times$ vBe | - | 0.014 | 0.861 | 0.511 |
| 5 | - | vBP/vBA $\times$ vBe | 0.538 | 0.860 | 0.508 |

The first variable entered into the model was TBA multiplied by CI$^2$, explaining 84% of yield variability. In the second step, the derived variable vBP/vBA×vBe was selected, increasing the $R^2$ by an additional 1.2% but later removed in step five by surpassing the 0.15 significance level for entry. In a third step, the derived variable vBA/vBP was chosen, which was followed, in the fourth step, by the variable vBA/vBP×vBe, achieving a final $R^2$ of 0.86. No other variable met the 0.15 significance level for entry into the model.

The estimated values of the elected model fit very well with the actual yield. However, the residual plot (not shown) indicated that the estimated yield variation was dependent on the values of the predictor variables. The violation of the constant variance assumption indicated the need for a variable transformation. A maximum likelihood analysis suggested

that a square root transformation of the response variable (yield) would be more appropriate than the original scale. The square root transformation stabilized the variance and improved the linearity of the model (data not shown). Equation (3) shows the final model after converting the rooted response back to the original scale.

$$\text{Est. yield}_{\text{image}}\ (\text{kg/m}) = (0.883 + 298\text{E} - 7\ \times\ [\text{TBA}\ \times\ \text{CI}^2] + 0.0004\ \times\ [\frac{\text{vBA}}{vBe}\ \times\ \text{vBP}] - 1.54\ \times\ \frac{\text{vBA}}{\text{vBP}})^2 \qquad (3)$$

Figure 6 shows the results of applying Equation (3) to both the training and validation sets, also on the original scale, as this is the scale that would be of interest to a user. The visual appraisal shows a good agreement between the observed and estimated yield on both data sets.

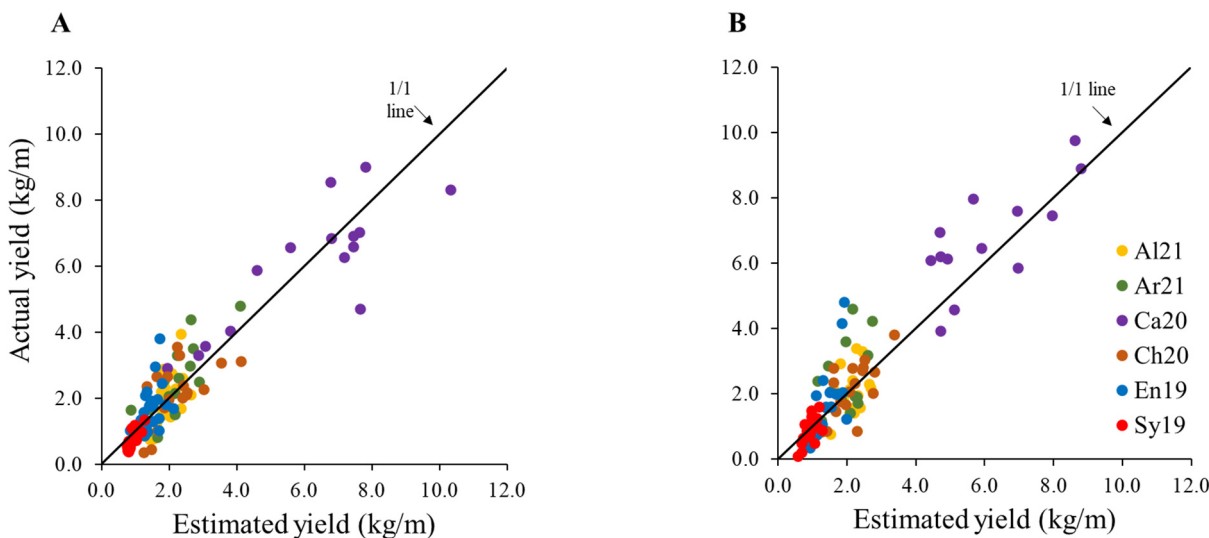

**Figure 6.** Regression analysis between the observed and estimated yield (obtained using Equation (3)) for the (**A**)) training (n = 108) and (**B**)) validation (n = 105) data sets, separated by cultivar.

Table 5 shows the statistical indicators of goodness of fit of the multiple regression model presented in Equation (3). The results corroborate the visual evaluation on both the training and validation sets, with a high and significant $R^2$ and a low RMSE. The overall fitted lines' slope and intercept did not show significant differences for 1 and 0, respectively, meaning that the model does not present an underestimation or overestimation trend. The $R^2$ decreased, and the RMSE increased slightly after the root transformation (only applicable to the training set).

**Table 5.** Statistical indicators of the model represented by Equation (3), on both the training and validation sets. All $R^2$ values are statistically significant ($p < 0.001$).

| Model | n | $R^2$ | Adj-$R^2$ | RMSE | Linear Regression | |
| | | | | | Intercept [1] | Slope [2] |
|---|---|---|---|---|---|---|
| Training set | 108 | 0.85 | 0.85 | 0.74 | 0.201 [ns] | 0.934 [ns] |
| Validation set | 105 | 0.86 | 0.86 | 0.82 | 0.019 [ns] | 1.082 [ns] |

[1] *t*-test for intercept = 0; [2] *t*-test for slope = 1; [ns] = not significant.

### 3.3. Comparing Manual with Image-Based Yield Estimations

In order to measure the effectiveness of the estimation method presented in this work, fruit mass estimations obtained using the stepwise regression model explained above were compared with manual yield estimations obtained via the conventional method based on bunch counts and historical bunch weight. Figure 7 shows both estimations paired with the

actual yield per meter on the validation set. The visual assessment shows a good fit between both the manual and image-based estimations with actual yield, with all lines presenting synchronized trends. In cvs. Chardonnay and Syrah, the manual yield estimation method presented a general overestimating trend. Both yield estimating methods presented the same overall correlation coefficient between estimated values and actual yield ($r = 0.89$); however, under and overestimations can be observed in both cases.

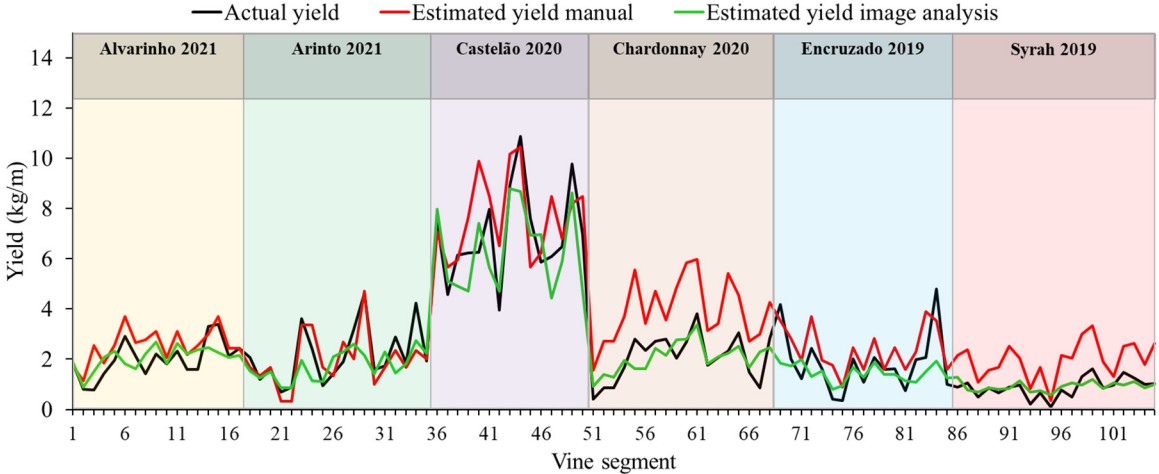

**Figure 7.** Variation of actual yield per vine segment and cultivar compared to estimated yield using manual and image-based methods.

Table 6 presents the accumulated yield estimations obtained on the validation set for every single cultivar. It includes the estimation error when compared to the actual yield. Regarding manual estimations, the lowest absolute error was observed on cv. Arinto followed by cv. Castelão, while the highest was observed on cv. Syrah. Only cv. Arinto presented an absolute estimation error below 10%, while cvs. Chardonnay and Syrah presented extremely high errors, close to twofold the actual value. When extrapolated to the whole vineyard plot, estimation errors changed considerably for most cultivars: 25.6%, −13.1%, 6.5%, 16.3%, 27.9%, 30.1% for cvs. Alvarinho, Arinto, Castelão, Chardonnay, Encruzado and Syrah, respectively. Image-based estimations presented lower absolute errors than manually obtained ones in five out of six datasets, with only two cultivars presenting values above |10%|. The image-based estimation with the lowest error was obtained on cv. Chardonnay, followed by cvs. Alvarinho and Syrah, all three cases with values below |10%|. The highest error was observed on cvs. Arinto and Encruzado. Overall, with all cultivars combined, the image-based method presented a lower absolute estimation error than the manual approach. For all cases, with the exception of cv. Castelão, the correlations between actual and estimated yield were higher for the manual method.

**Table 6.** Results of estimated yield obtained using manual and image-based methods compared to actual yield obtained at harvest, on the validation set, for each cv and season. Error = (Yield$_{est}$ − Yield$_{act}$)/Yield$_{act}$; r: Pearson correlation coefficient between estimated and actual yield.

| Cultivar | N | Actual Yield | Estimated Yield $_{manual}$ | | | Estimated Yield $_{image}$ | | |
|---|---|---|---|---|---|---|---|---|
| | | (kg/m) | (kg/m) | Error (%) | r | (kg/m) | Error (%) | r |
| Al21 | 17 | 2.0 ± 0.1 | 2.6 ± 0.1 | 27.5 | 0.73 | 2.1 ± 0.1 | 2.8 | 0.51 |
| Ar21 | 18 | 2.1 ± 0.2 | 2.0 ± 0.2 | −7.6 | 0.81 | 1.7 ± 0.1 | −19.0 | 0.64 |
| Ca20 | 15 | 7.0 ± 0.3 | 7.7 ± 0.3 | 10.0 | 0.62 | 6.4 ± 0.3 | −9.2 | 0.75 |
| Ch20 | 18 | 2.1 ± 0.2 | 3.9 ± 0.2 | 90.0 | 0.83 | 2.1 ± 0.1 | −0.1 | 0.74 |
| En19 | 17 | 1.8 ± 0.2 | 2.4 ± 0.1 | 29.9 | 0.79 | 1.5 ± 0.1 | −20.8 | 0.67 |
| Sy19 | 20 | 0.9 ± 0.1 | 2.0 ± 0.1 | 124.2 | 0.86 | 0.9 ± 0.0 | 4.4 | 0.65 |
| *Pooled* | 105 | 2.5 ± 0.1 | 3.3 ± 0.2 | 31.0 | 0.89 | 2.3 ± 0.1 | −8.3 | 0.89 |

## 4. Discussion

The present work compares the use of grapevine image features with a manual method based on bunch counts and historical bunch weight to estimate vineyard yield non-invasively near the veraison phenological stage. The manual method is obtained by multiplying bunch counts with historical bunch weight, while the image-based method builds on previous publications [42], where the total bunch projected area is estimated from canopy porosity and visible bunch area. A new model was computed in order to accurately convert the output of that work into vine yield.

### 4.1. Yield Estimators

The dataset used in this work, which encompasses six subsets, corresponding to six cvs. collected in three vineyard plots, presents significant variability, with similar previously reported magnitudes [39], an important feature required to validate our hypothesis. Previous research reports that bunch number explains approximately 60–70% of the seasonal variation in vine yield [8]. In general, our results confirm this trend, with the $R^2$ between bunch number and vine yield varying between 0.56 and 0.74 (Table 2). This highlights the fact that spatial variability regarding bunch number can have an important effect on the output of a manual estimation based on bunch count samples that are extrapolated to the whole plot. In the present work, because manual estimations were performed using the actual bunch number, the errors showed by this approach should be attributed to the variability of the bunch size between the analyzed vine segments combined with the difference between historical and actual bunch weight. In our data, the historical bunch weight was considerably different from the actual bunch weight in several cultivars, which hints at potential estimation errors (Table 2). Although this manual method is a very accessible approach that has the advantage of being possible to obtain at a very early stage, the fact that it is so dependent on historical data makes it unreliable in many scenarios, as has been previously pointed out by other authors [3,50]. It is important to highlight the particularly low yield presented by the cv. Syrah in the studied conditions (Table 2), which can mainly be attributed to its low average bunch weight, is almost half of the historical data. Such differences were potentially caused by the joint effect of poor fruit set conditions [51] and some berry shriveling near harvest (visually assessed) commonly observed in this cultivar (e.g., [52]). Cultivar Syrah vines, in the studied vineyard plot, are also characterized by a considerable age (~26 years old), which may have affected manual estimations. Regardless, it is important that in these types of vineyard plots, which are likely to be more variable than younger ones, yield estimation methods are also accurate, particularly in Europe, where an estimated 37.1% of total planted vines are older than 30 years [53].

From an image-based perspective, regarding visible bunch area, the highest values observed on cvs. Castelão and Chardonnay (Table 3), from the QA site, can be explained not only by its higher yield but mostly by the fact that, on this site, basal defoliation was performed at the pea-size phenological stage. This canopy management practice is very common in coastal areas, aiming at reducing canopy density in the bunch zone for bunch health and berry color purposes [51]. Even though lateral shoots compensated part of that defoliation by harvest, at veraison, bunches still presented high exposure. This is further confirmed by the fact that the difference between cultivars regarding their estimated TBA is not as large, highlighting the robustness of previously presented approaches [42] to consider different scenarios of bunch exposure. Moreover, although cv. Encruzado presented the lowest BA of all cvs., because of its low POR, estimated TBA was still higher than cv. Syrah and close to other cultivars, indicating that a low percentage of visible bunches was estimated [42]. In cv. Syrah, the 142.7 cm$^2$/m of BA most likely represents a high percentage of the total bunch area, being a case with very low yield and high POR. The higher porosity presented by cv. Syrah can be explained by the low vigor and mild water stress (rainfed vineyard) that was observed in this vineyard. This cultivar showed some leaf senescence, which was already visible at veraison and had an impact on POR and, consequently, on bunch exposure [54]. Moreover, cv. Syrah also showed a very high average

vBe value (Table 3). This can be explained by this cultivar's lower CI when compared to the remaining ones. Although some of the cultivars ended up presenting denser bunches near harvest, as a factor of their berry growth during maturation, at veraison, most presented a CI = 5. However, cv. Castelão was already more compact than the other cultivars, probably as a consequence of this cultivar's larger berries. In fact, by dividing vBA by vBe, this can be observed. Although this ratio is not the true value of the average berry area, as both variables consider partially visible berries, it can be seen as an indicator of that trait. Furthermore, cv. Syrah, which was the only cultivar with a CI = 3 at veraison, also presented the lowest vBA/vBe ratio; this is probably what caused its bunches to be so light. With smaller berries and looser bunches (Table 3), it was possible to see a higher percentage of berries of the same bunch than in other cases.

Regarding bunch perimeter, although closely related to BA in terms of bunch imaging characterization, it presents some noticeable differences in some cases. Even though cv. Alvarinho presented higher relative values of BA than BP; this was not true for any other cultivar (Table 3). Cv. Alvarinho presented a high number of very small bunches (Table 2), a fact that has an impact on their distribution along the canopy. Consequently, this cultivar's vine images presented a higher number of segmented bunch pixel agglomerates, i.e., higher perimeter for the same bunch area than a cultivar with larger bunches (e.g., cv. Arinto). Furthermore, although vBe presented a very high correlation coefficient with vBA (r = 0.82), they have been proven to not be redundant when used to estimate bunch weight [25,45]. These variables were not used singularly in the present work. However, their ratio provides us with information regarding the average berry size. This can be valuable when attempting to evaluate bunch architecture as it is dependent on berry size and can have an impact on bunch weight [47].

In the present work, images from two different imaging systems were used: on-the-go, using the Vinbot platform, and static, using a commercial camera. As image analysis was performed manually, image quality was not relevant to obtaining image traits with accuracy. However, the time taken to analyze lower resolution images (Vinbot) was higher, as the difficulty increased with lower image quality. This can be a challenge when analyzing data automatically, as classification algorithms might not achieve the same results on both types of images. The variable that was most sensible to lower resolution seemed to be vBe, as it was harder to differentiate berries from one another in shaded parts of the image. Higher resolution images will always perform better in terms of data quality, with the inconvenience of potentially causing higher processing times as a consequence of larger images [55].

### 4.2. Conversion of Image Traits into Mass

The exponential relationship presented between TBA and actual yield (Figure 5B) indicates that TBA represents higher relative yield at higher magnitudes, i.e., the rate of change of yield depends on TBA itself. The relatively high residuals could either be a consequence of the original TBA estimation, which is based on BA and POR and encompasses some error ($R^2 = 0.80$, [42]), or the relationship between TBA and bunch mass. As the true TBA is unknown, it is not possible to know with certainty if the error originated when estimating it. However, in the original publication, where the true TBA is known [42], specific results for the cv. Syrah showed overestimations of TBA, which is also potentially true in the present case, as the cv. Syrah observations are almost all below the regression line. Furthermore, the direct relationship between the true TBA and yield may be subject to variation caused by bunch occlusions by other bunches, which have a higher chance to occur at higher yields, such as the case of cv. Castelão. In fact, in Figure 5B, we can observe that there are almost no observations below the regression line (overestimations) above yields of ~5 kg/m, which coincides exclusively with cv. Castelão. Because of this, it is unclear if this behavior is characteristic of high-yielding vines or a trait of this cultivar. Moreover, cv. Encruzado was the only one to show an r between TBA and actual yield below 0.50 (Figure 5A). This cv. was also the one with the lowest vBA and POR while still

having an average actual yield of 1.8 kg/m. It can be assumed that cv. Encruzado presented a particularly challenging scenario, with a very low percentage of visible vBA compared to its true TBA. Considering this, there is a possibility that the yield estimation error found in Figure 5B is a consequence of an accumulated error from both steps (TBA estimation and area to mass conversion) and that this error is cultivar-dependent. These results justify the strategy of adding other variables in order to evaluate if the model accuracy improves at estimating vine yield.

As a reliable estimation of bunch compactness was not possible to achieve with the analyzed variables, the OIV bunch compactness index was added to the model. Bunch compactness obtained via OIV methodology [46] must be performed visually on a sample of bunches. However, this can be performed non-invasively, on the vine images themselves, by an expert who is running the yield estimation system (i.e., the service provider). Furthermore, it is not necessary to obtain this CI from a large number of bunches as it was used merely as an indicator of how compact bunches are for that cultivar plot. It could even be obtained historically or from OIV records [46] if the cultivar is included on their list. However, it is always advised for bunch compactness to be estimated on bunches from the plot being assessed, especially when estimating yield at veraison, as CI in the bibliography refers to fully matured bunches. There are some upgrades that can be devised in the future regarding this topic. For example, if the resolution is high enough, or images are collected closer to the vines, then part of the previously reported results [45,48] could possibly be mirrored. If so, CI could be automatically obtained on every image, which could potentially improve the yield estimation model, as bunch compactness can vary even within the same cultivar.

Following the stepwise regression analysis, the variables selected for the image-based approach (Table 4) seem to be effective at decreasing the error generated by the problems mentioned above. In the first step of this analysis, the variable TBA was selected and associated with CI, corroborating previous work [42] and validating the importance of this variable in explaining grapevine yield. It also adds the importance of the variable CI in differentiating TBA between cultivars with different bunch architecture, as bunch compactness is known to be a trait that influences bunch weight [47]. This trait is highly variable between cultivars [56] and can be one of the reasons why it is challenging to estimate yield with the same method across different cultivars. In Victorino et al. [45], the authors explore this in laboratory conditions by using a ratio between vBA and vBP, as a bunch with a higher percentage of holes in its structure will show a lower ratio.

Furthermore, in Cubero et al. [48], the authors also explore the estimation of bunch compactness using automated image analysis, recurring to variables such as bunch length, perimeter and area. However, such approaches are much harder to achieve at the vine level. The main limitations being image resolution and noise. While in lab conditions, differences between berries and blank spaces inside the bunch are clear, within a vine image, this is not as easy, as the resolution does not allow it. Furthermore, the noise behind bunches also makes it harder for blank spaces within the bunch to be visually identified in an image by a human and probably also by an automatic algorithm. However, as mentioned above and shown by the third step of the stepwise regression model (Table 4), the ratio between vBA and vBP can still be very valuable even at the vine level. In fact, as explained above, this ratio can be seen as an index that distinguishes a bunch pixel agglomerate's size. In the images, this size can be affected not only by the above-mentioned bunch architecture but also by factors such as bunch proximity to one another and even leaf occlusions. That is, in the image, two bunches that are side by side will hardly be distinguished and instead will be counted as one larger bunch (lower perimeter for the same area). On the other hand, a leaf occlusion that segments, in the image, a large bunch in two will simulate two smaller bunches instead of one larger (higher perimeter for the same area). Furthermore, this ratio can also be an indicator of bunch format, as the ratio between vBA and vBP is maximum within a circle. Although it is highly unlikely that a bunch pixel agglomerate resembles

a perfect circle, this ratio may distinguish cases that present rounder shapes over more angular ones.

The final derived variable to enter the model (vBA/vBP $\times$ vBe) distinguishes cases with constant vBA/vBP, by associating it with an indicator of bunch size. That is, it adds the possibility to differentiate cases with the same type of bunch pixel agglomerates, considering its shape and distribution in the image, but with different potential yield (higher number of visible berries versus lower one). Furthermore, the fact that vBe was considered by the model, even if as part of a derived variable, is in accordance with previous publications that elected vBe as an important estimator of yield (e.g., [26,27,29]). However, as previously reported [25,45], accuracy increases when used along with other predictors.

*4.3. Comparing Manual with Image-Based Yield Estimations*

The fact that the image-based yield estimation method achieved results that rival and, in most cases, outperform the ones presented by the studied manual method (Figure 6 and Table 6) is highly promising. Indeed, only two image-based cases showed estimation errors above the |10|% error threshold mentioned for near-harvest estimations in [15], compared to four cases when using the manual approach. Three image-based cases (cvs. Alvarinho, Chardonnay and Syrah) even showed an error below the strictest reported threshold of 5% indicated by [4]. Even though some errors were found on individual observations on the training and validation sets (Figure 6), it seems that most under and overestimations compensated for each other, achieving overall good results on the combined set of vines in the validation set. Although yield estimation results are generally good, the fact that some cultivars present significantly higher estimation errors is an indicator of cultivar dependence. However, the highest errors were obtained on the same cultivars where the densest canopies were found (cvs. Encruzado and Arinto with the lowest vBA and POR; Table 3). These results hint that grapevines with extremely dense canopies can be a limitation of the proposed yield estimation approach. On the other hand, high vigor scenarios can be affected by factors such as cultivar, overall vineyard management and growing conditions [9]. Regardless, besides a lower overall yield estimation accuracy, values obtained with the image-based approach followed actual vine yield trends very closely for these and the remaining cultivars (Figure 7). This is relevant if we consider the scale at which this information is important. That is, having good individual estimations of every vine is important to evaluate the quality of this approach; however, if the purpose of the yield estimation performed by a winegrower is to, for example, prepare the cellar to receive grapes, then maybe such a high resolution is not required. Instead, a broader estimation of a group of vines is enough. If the goal is to provide relevant information for marketing strategies, then even wider estimations of cultivar plots would suffice. On the other hand, if the winegrower is planning yield regulating tasks, then possibly, plant by plant information can be useful. The present results show that the image-based method studied in this work can be effective in both scenarios for almost all cases (Figure 7; Table 6).

The manual method explored in this work is highly accessible to any winegrower, and it is possible to perform even before fruit set when inflorescences and the following bunches remain the same until harvest. However, in some cases, the method showed extremely high estimation errors, mainly caused by differences between historical and actual bunch weight, which, in some cultivars, was almost twofold (Table 2). This was particularly true for cv. Syrah, where the considerably higher historical bunch weight caused a significant overestimation of yield on the validation set. As the image-based method is not dependent on historical data, it was a case where the approach designed in this work was highly advantageous. However, the overestimation was presented when using the manual method on the cv. Syrah, and also on cv. Chardonnay was significantly lower when data were extrapolated to the whole cultivar plots (error = 30% and 16%, respectively, for cvs. Syrah and Chardonnay). This is most likely related to the spatial variability that the plot presents and the possibility that images were collected and sampling was performed on vines that were less representative of the plot. In fact, the number of

plants in the validation set represents only 1% to 4% of the total number of plants in the cultivar plots. This further proves the difficulty of a sample-based approach to be effective when spatial variability is present [57]. Even though, in this case, the error decreased from the validation set to the whole plot extrapolation, for the manual approach, it was a consequence of the spatial variability compensating for the errors caused by season variability (historical bunch weight). Furthermore, this extrapolation exercise would be unrealistic to perform for the image-based yield estimation approach as one of the most relevant advantages of an automated method is the fact that it can be used on a high number of vines. As such, an automated image-based setup (e.g., [28]) would be possible to use on all vines of the cultivar plots. Such an approach would leave out the need for extrapolation, and the errors of such an approach would be closer to the ones presented in Table 6.

Further research includes adding more vine traits that can be easily scanned using image analysis (e.g., average berry size, vine leaf area) or by adding other technologies that collect data beyond what can be obtained from an RGB image [18,24,41]. In an optimal scenario, the data collected from each vine could be carried to future seasons, i.e., developing a model that considers detailed historical data collected with the same image collection device, similar to what has been explored for enhanced conventional sampling strategies [14]. Furthermore, the collection of larger datasets of different cultivars in different growing conditions can be important to increase the model robustness to be applied across different scenarios. This can potentially enable the use of models developed for specific cultivars or growing conditions. Moreover, further work is required to improve estimations of TBA on very dense canopies with high percentages of bunch occlusion by leaves. Further, as previously mentioned, bunch occlusion by other bunches can play an important role in an image-based setup. We are currently working on this topic and exploring what vine traits are connected to this issue, for example, fruit zone amplitude, bunch number and bunch distance to other bunches. Moreover, we are also exploring if analyzing images from both sides of the canopy can be advantageous regarding this and other topics.

## 5. Conclusions

In this study, a non-invasive, image-based yield estimation method was presented and compared against estimations performed using a manual method based on bunch counts combined with historical bunch weight at harvest.

Total bunch projected area, estimated from bunch area and canopy porosity, although successful at estimating part of the occluded bunch pixels, did not allow an accurate yield estimation across multiple cultivars at the vine level. Other image-based variables, such as the visible berry number, bunch perimeter and bunch compactness index, were added as derived variables, which highly increased the accuracy of the model. The final model achieved accurate results in both individual vines and when analyzed in bulk, which can be important depending on the goal of the yield prediction.

Overall, the yield estimation results based on the image-based approach suggested in this work rivaled the ones obtained with the manual method, in most cases even outperforming them. Although the manual method is accessible to any winegrower and obtainable at earlier phenological stages, the studied image-based approach presents the advantage of potentially being fully automated and used across whole vineyards non-invasively. Further work is required to fully automate this methodology and to explore additional easily obtained variables that may improve final estimations and robustness over different vineyard scenarios.

**Author Contributions:** Conceptualization, G.V., R.P.B., J.S.-V. and C.M.L.; methodology, G.V., R.P.B., J.S.-V. and C.M.L.; software G.V. and C.M.L.; validation, G.V., R.P.B., J.S.-V. and C.M.L.; formal analysis, G.V. and C.M.L.; investigation, G.V.; resources, G.V., R.P.B., J.S.-V. and C.M.L.; data curation, G.V.; writing—original draft preparation, G.V.; writing—review and editing, G.V., R.P.B., J.S.-V. and C.M.L.; visualization, G.V. and C.M.L.; supervision, R.P.B., J.S.-V. and C.M.L.; project administration, C.M.L.; funding acquisition, G.V. All authors have read and agreed to the published version of the manuscript.

**Funding:** Gonçalo Victorino was supported by the PhD grant SFRH/BD/132305/2017, sponsored by FCT—Fundação para a Ciência e a Tecnologia.

**Data Availability Statement:** Not applicable.

**Acknowledgments:** We gratefully acknowledge the valuable contributions of masters students Guilherme Maia, José Queiroz, Beatrice Carmignani, Giuseppe Samà, Luigi Mauro, Ruben Bonaria, João Mak, Nuno Simão and Marta Simões who helped with data collection and overall teamwork that made this research possible.

**Conflicts of Interest:** The authors declare no conflict of interest.

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
