# Peer review of "Comparing a New Non-Invasive Vineyard Yield Estimation Approach Based on Image Analysis with Manual Sample-Based Methods"

_agronomy, doi:10.3390/agronomy12061464_

Round 1

Reviewer 1 Report

This study is fascinating, offering a new approach to estimating vineyard yield using a camera that claimed to be better than the manual method. The paper was well-written and scientifically sound. However, some remarks needed to be confirmed/ added to increase the clarity for the reader about the study. I have some recommendations to add/change as follows:

·        Line 73-80: The existing method and references must be described. Recommend adding a table to compare the pros and cons of the existing method.

·        Line 146-149: Recommend adding a figure/ a map to show the locations.

·        Line 185-187: Need to be explained in the main text.

·        Line 230: Can you add a small explanation about the previous method?

·        Line 273-275: Why are you splitting 50:50? not 60:40 or 70:30?

·        Line 438: Can you add some explanation on the "manual" approach in the method section.

·        Line 502: Splitting the paragraph into two paragraphs.

·        Line 554: Splitting the paragraph into two paragraphs.

·        Line 584: Splitting the paragraph into two paragraphs.

·        Line 683-684: it is not "very" accurate. Probably, acceptable accurate?

Reviewer 2 Report

Reviewing Comments for Manuscript Agronomy-1751789

Victorino et al. presented a study that developed a new image analysis-based vineyard yield estimation approach and evaluated the accuracy of this new approach by comparing it with the conventional manual sample-based method. The authors obtained data from three vineyard plots in Portugal, covering six cultivars in three growing seasons. Based on created variables, the authors applied a stepwise regression to predict the yield. The proposed method achieved a good R2 of 0.86 in validation, indicating that the new approach outperformed the conventional manual method. The experiment is well designed. And the conclusions are supported by results and discussion. Therefore, I recommend a minor revision. And my major comments are as follows:

1. The authors applied a stepwise regression to predict yield. However, detailed information about the stepwise regression is not provided in the manuscript, such as variable selection criteria and performed steps. Please add the necessary information.

2. According to regression results, the accuracy of provided method seems to be cultivar-depended. For example, Ca20 and Sy19 may have a better performance as compared with En19. However, it is not possible to conduct regression for each cultivar due to the limitation of the sample size. Do you have a solution for this observation? If not, please discuss it in the limitation related context.

 3. When conducting the regression, the authors equally split all data into training and validation sets. What if using five-fold cross-validation? On the other hand, the relative sample size for regression is another limitation of the study. Please provide a discussion.

Reviewer 3 Report

Dear Authors

The present work aims at comparing the accuracy of a new non-invasive and multi cultivars, image-based yield estimation approach with a manual conventional method. Data was collected from six cultivars for three seasons from three vineyard plots, in Portugal.  A stepwise regression model was used to select the most appropriate set of variables to predict yield. A combination of derived variables was obtained that included visible bunch area, estimated total bunch area, perimeter, visible berry number and bunch compactness. The model achieved a R2 = 0.86 on the validation set. 

- Line 15: please rewrite more clearly “Images were obtained from 213 non-disturbed one-meter vine segments at the beginning of veraison”.

- Lines 31-33: Add the following sentence “It can help plan strategies of grape thinning and vineyard in general management as a yield regulation practice, or to plan harvest in terms of logistics, scheduling, workforce, and machinery requirements [1-2]”.

[1] Cataldo, E., Salvi, L., Paoli, F., Fucile, M., & Mattii, G. B. (2021). Effects of defoliation at fruit Set on vine physiology and berry composition in cabernet sauvignon grapevines. Plants, 10(6), 1183.

[2] Aru, V., Nittnaus, A. P., Sørensen, K. M., Engelsen, S. B., & Toldam-Andersen, T. B. (2022). Effects of Water Stress, Defoliation and Crop Thinning on Vitis vinifera L. cv. Solaris: Part I: Plant Responses, Fruit Development and Fruit Quality. Metabolites, 12(4), 363.
